# Is Hysteroscopy Prior to IVF Associated with an Increased Probability of Live Births in Patients with Normal Transvaginal Scan Findings after Their First Failed IVF Trial?

**DOI:** 10.3390/jcm11051217

**Published:** 2022-02-24

**Authors:** Athanasios Zikopoulos, Apostolia Galani, Charalampos Siristatidis, Ioannis Georgiou, Eirini Mastora, Maria Paraskevaidi, Konstantinos Zikopoulos, Efstratios Kolibianakis

**Affiliations:** 1Department of Obstetrics and Gynaecology, Royal Cornwall Hospital, Truro TR1 3LJ, UK; 2Department of Obstetrics and Gynaecology, Royal Preston Hospital, Preston PR2 9HT, UK; liagalani90@gmail.com; 3Assisted Reproduction Unit, Second Department of Obstetrics and Gynecology, Faculty of Medicine, National and Kapodistrian University of Athens, 11528 Athens, Greece; harrysiris@gmail.com; 4Assisted Reproduction Unit, Department of Obstetrics and Gynaecology, Faculty of Medicine, University of Ioannina, 45500 Ioannina, Greece; igeorgio@uoi.gr (I.G.); eirinimastora1@gmail.com (E.M.); kzikop22@gmail.com (K.Z.); 5Department of Metabolism, Digestion and Reproduction, Institute of Reproductive and Developmental Biology, Faculty of Medicine, Imperial College London, London W12 0HS, UK; m.paraskevaidi@imperial.ac.uk; 6Unit of Human Reproduction, First Department of Obstetrics and Gynaecology, Faculty of Medicine, Aristotle University of Thessaloniki, 56403 Thessaloniki, Greece; stratis.kolibianakis@gmail.com

**Keywords:** assisted reproduction technology, in vitro fertilisation, intracytoplasmic sperm injection, hysteroscopy

## Abstract

(1) Background: Nowadays, pregnancy can be achieved by in vitro fertilisation (IVF) or by intracytoplasmic sperm injection (ICSI) for many infertile couples. However, implantation failure still remains a significant problem and it can be stressful for both patients and doctors. One of the key players for pregnancy achievement is the uterine environment. Hysteroscopy is the most reliable method to evaluate the uterine cavity and to identify any intauterine pathology. The aim of this retrospective study was to compare live birth ranges in between women who after a first failed IVF/ICSI attempt underwent a hysteroscopy and those who were evaluated by a transvaginal scan. (2) The retrospective study took place at the Assisted Reproductive Unit of the University Hospital of Ioannina, Greece, from 2017 to 2020. It included 334 women with normal findings in a repeat ultrasound scan after a failed IVF/ICSI trial, 137 of whom underwent in turn diagnostic hysteroscopy before the next IVF/ICSI. (3) Results: Live birth rates were higher in the study group (58/137 vs. 52/197 *p* = 0.0025). Abnormal endometrial findings were identified in 30% of the patients of the study group. (4) Conclusions: The addition of hysteroscopy as an additional investigation to those patients with a first failed IVF/ICSI could improve the rates of live births. A properly conducted RCT could lead to a robust answer.

## 1. Introduction

Infertility remains a significantly stressful matter for many individuals. During the past few decades science has developed ways to handle infertility such as in vitro fertilisation (IVF) and intra-cytoplasmic sperm injection (ICSI). However, implantation failure after IVF or ICSI can be quite frustrating for both clinicians and patients. The probability of a pregnancy achievement is approximately 30% [1], while implantation failure may often be associated with uterine cavity abnormalities, such as endometrial polyps, small submucous fibroids, adhesions, and septa [1]. There is evidence to suggest that these may exert a negative impact on the chance to conceive through IVF or ICSI.

Methods for assessing the endometrial cavity include transvaginal sonography (TVS), hysterosalpingography, saline infusion sonogram and hysteroscopy (HSC) [2]. From the aforementioned options, hysterosalpingography is characterised by a low specificity and high false-negative and false-positive rates [3]. The diagnostic accuracy of the saline infusion sonogram is high; however, it is an examination which can be quite uncomfortable as an office procedure, specifically for nulliparous women [4]. On the other hand, transvaginal sonography is a non-invasive and reproducible technique; however, it is not very sensitive [3]. HSC is currently the only method for directly observing physiological and pathological changes in the endometrium that allows targeted biopsies to be performed and specific treatments to be applied [5]. It has been reported that the prevalence of minor intrauterine abnormalities identified by HSC is as high as 30 to 45% compared to normal transvaginal sonography, and abnormalities found by HSC are significantly higher in patients with previous assisted reproductive technique failure [6,7].

Implantation failure after IVF/ICSI refers to the absence of implantation after good-quality transfer. It can be related either to maternal factors or embryonic reasons [7]. The most common maternal causes are uterine abnormalities, endocrine disturbances, thrombophilic and immunological causes [8]. In this study, the role of the uterine abnormalities was specifically evaluated. The purpose of this retrospective study was to evaluate whether, in patients with one IVF failure and normal transvaginal scan findings, hysteroscopy prior to the next IVF/ICSI trial is associated with an increased probability of live birth.

## 2. Materials and Methods

This retrospective study took place at the Assisted Reproduction Unit of the University Hospital of Ioannina from January 2018 to December 2020. All patients were diagnosed with infertility. They all met the criteria for undergoing a controlled ovarian hyperstimulation protocol followed by IVF/ICSI. This trial protocol was approved by the Scientific Board and Bioethnics Committee of the University Hospital of Ioannina (Approval Number: 33800/14 November 2018). The goal of the study was to evaluate whether in patients with one IVF/ICSI failure and normal transvaginal scan findings, a hysteroscopy prior to the next IVF/ICSI is associated with an increased probability of live birth.

### 2.1. Patient Population/Eligibility Criteria and Study Design

The inclusion criteria for the study entry were as follows: history of a failed first IVF/ICSI attempt with high quality embryos (high quality embryos are those defined as good quality embryos on day 3 and those defined as good morphology blastocysts grade A and B) and normal TVS findings; BMI less than 30 kg/m^2^, age less than 43 years old, no substance abuse, absence of any known and/or untreated haematological or immunological disorders. Exclusion criteria included history of lower abdominal or pelvic infection, a higher chance of intra-abdominal infection due to intestinal surgery, endometriosis grade 3 and 4, previous caesarean section with niche formation, presence of untreated unilateral or bilateral hydrosalpinx, previous endometrial scratching, meno-metrorrhagia and untreated endocrine abnormalities. Regarding male factors, patients whose partner was diagnosed with azoospermia were also not included.

The study group included 137 women, while the control group included 197 women. Both HSC and TVS were performed in the early proliferative phase of the menstrual period (day 3–9). A 4.3 mm continuous rigid scope with a 30-degree view and normal saline as distention media were used. The endocervical canal, uterine cavity, tubal ostiums and endometrium were inspected methodically and the findings were recorded in a standardised form. In those cases where endometritis was suspected, hysteroscopy was combined with endometrial biopsy. The control group consisted of the patients who had an unsuccessful first attempt of IVF/ICSI and a repeat TVS before the second attempt.

### 2.2. Ovarian Stimulation Protocol

Ovarian stimulation was performed for all women using GnRH antagonists and recombinant Follicle Stimulation Hormone (FSH) or purified FSH 150–300 IU daily. Ovulation induction with hCG was achieved using 250 mcg choriogonadotrophin alfa, when at least three 17 mm follicles were seen on ultrasound scan. Ultrasound-guided oocyte retrieval was performed 36 h after final oocyte maturation. Embryo transfer was performed on day 3 with a soft catheter. Vaginal progesterone supplementation was used for luteal phase support and continued for up to 4 weeks after embryo transfer in the presence of a positive pregnancy test. The pregnancy test was carried out two weeks after oocyte retrieval and in those cases of pregnancy a TVS was performed on weeks 7 and 12.

### 2.3. Selection of Embryos and Embryo Transfer in a Fresh Cycle

Selection of embryos for transfer based on their morphological characteristics and their developmental rate. Regarding the quality, it was evaluated according to morphological criteria based on the number of blastomere, size, appearance.

### 2.4. Outcome Measures-Statistical Analysis Type

The primary outcome measure was live birth. Abnormal hysteroscopic findings, clinical pregnancy and miscarriage rates were the secondary outcomes. Abnormal hysteroscopic findings were also documented and treated accordingly either during the hysteroscopy when appropriate, such as in cases of endometrial polyps or after the hysteroscopy, such as in cases of endometritis.

Descriptive data were presented in (%) for qualitative data. Moreover, the data were assessed using the Generalized Lineal Model (GLM). Data in Table 1 were analysed using *t*-test and Chi-squared.

## 3. Results

### 3.1. Study Characteristics

The initial number of participants that were included in this study were 360, however, because of the failure to follow up and due to withdrawals, the final number of participants was 334. From those 26 patients that were excluded, 10 were from the study group and 16 from the control group. The study group consisted of 137 patients and the control group consisted of 197 patients.

Table 1 contains the main patients’ demographic and baseline characteristics.

### 3.2. Hysteroscopical Findings

The corrected hysteroscopical findings and the live pregnancy outcomes are shown in Table 2. Forty-one (30%) of the patients who underwent a hysteroscopy had abnormal hysteroscopic findings. Whenever the pathology could be corrected during the hysteroscopy (which was the case in instances such as endometrial polyps, while in some cases this was not possible, such as endometritis), treatment took place after the hysteroscopy. Specifically, 19 patients had a polypectomy for endometrial polyp, 13 had findings of submucosal fibroid—11/13 of them had a myosure resection of the fibroid while the 2 had a myomectomy using versa point. Four patients had a finding of adhesions and were treated with resection. Endocervical polyp was the hysteroscopic finding for 2 patients and both underwent a polypectomy. Endometritis was the finding for 3 patients and were treated with doxycycline and metronidazole.

### 3.3. Analysis of Primary and Secondary Outcomes

Proportion comparisons for the endpoints are shown in Table 3. The live birth comparison returned significant results (*p*-value 0.0025). The secondary comparisons returned nonsignificant results.

## 4. Discussion

The aim of this study was to evaluate whether hysteroscopy in the cycle prior to a second attempt of IVF/ICSI can lead to a higher probability of pregnancy compared to having no hysteroscopy, in women with a failed first attempt and normal findings at TVS. While other studies have chosen to study women with recurrent implantation failure (RIF)-study Siri [9], this study focuses on women who have failed to achieve pregnancy after a first unsuccessful IVF/ICSI attempt in order to potentially increase the efficiency of Assisted Reproductive Technology (ART) earlier in the course of treatment, especially when considering how stressful and financially demanding IVF/ICS can be.

At present, there is no high-quality evidence to support the standard use of hysteroscopy as a screening tool before IVF/ICSI. Although other imaging modalities, such as hysterosalpingogram or a transvaginal scan, are easy to perform, hysteroscopy allows a more accurate visual assessment of the endometrial cavity and offers the possibility of performing therapeutic interventions where appropriate [10]. The two main issues against the hysteroscopy use are its invasive nature and the uncertainty regarding the clinical significance of the observed intrauterine pathology on fertility [10,11]. The European Society of Human reproduction and Embryology (ESHRE) guidelines indicate hysteroscopy to be unnecessary, unless it is for the confirmation and treatment of doubtful intrauterine pathology [12]. However, it should not be said that HSC is a minimally invasive procedure, with a very low technical failure rate that can be performed as an outpatient procedure with no need for hospitalisation or anaesthesia [13]. This study has showed that women who are scheduled for their second IVF/ICSI attempt after having a hysteroscopy can achieve higher pregnancy rates.

A positive effect of hysteroscopy on the outcome of in vitro fertilization has also been shown previously, suggesting that a diagnostic hysteroscopy should be performed before expensive procedures such as assisted reproduction [14,15]. In that study, 21.1% of patients had confirmed abnormalities that had to be treated before performing IVF/ICSI. In 2014, a meta-analysis showed increased live birth rates after hysteroscopy in women scheduled for a first IVF cycle [16]. However, other investigators showed a benefit of routine hysteroscopy only in women 40 years and older [17].

It is likely that a beneficial effect of hysteroscopy is present if the proportion of women studies have an identifiable pathology at HSC. In this respect, it has been shown that women aged > 40 years old have a high probability of endometrial pathology, such as submucous myoma, endometrial hyperplasia, and polyps [18]. In this respect, they may represent a target group. A similar finding has been shown in women > 35 years of age [19]. On the contrary, it has been suggested in the TROPHY trial that no improvement in live birth rates are present after hysteroscopy in women with 2 to 4 failed IVF cycles. In another randomized trial that enrolled 750 patients scheduled for their first IVF cycle, hysteroscopy did not improve the live birth rates in women with a normal transvaginal ultrasound [20].

The higher pregnancy rate in women who were examined with hysteroscopy before IVF/embryo transfer (ET) cycles, as compared to women without undergoing one, was also reported back in 2005 (2005). However, no consensus has been reached on whether hysteroscopy may be able to act as part of a routine diagnostic screening process before starting IVF/ET programs [21].

Finally, as it is obvious by the *p*-values in Table 1 that the two groups were quite different with regards to age, BMI, duration of infertility and the number of the oocytes that were retrieved. All of these factors could affect either the endometrial pathology or the scan imaging (i.e., a high BMI can make difficult to have proper ultrasound images) and subsequently affect the results. Therefore, in the future, a well-designed randomized controlled trial (RCT) should be performed to provide precise answers to this clinical question. 

## 5. Conclusions

The current study suggests that screening hysteroscopy may increase clinical pregnancy rates compared to no intervention in women undergoing a second IVF/ICSI attempt with normal TVS findings. Outpatient or a hysteroscopy under anaesthesia, may become a recommended step for infertility workup before IVF/ICSI, even with normal TVS findings. However, a well-designed RCT should be performed with adequately described randomization and allocation concealment methods to provide a robust answer to this clinical question.

## Figures and Tables

**Table 1 jcm-11-01217-t001:** Main Study Characteristics.

	Group HSC	Group TVS	Means of 2 Groups—*p* Value
No. of patients	137	197	-
No. of cycles	137	197	-
Age (years)	35 (28–42)	36.5 (27–43)	1.41 × 10^−15^
Mean BMI (kg/m^2^)	27.4 (23–31.8)	28.8 (24–33.6)	8.9 × 10^−15^
Duration of infertility (years trying to conceive)	4 (2–6)	3 (2–4)	2.2 × 10^−16^
Causes of infertility in numbers	-PCOS (*n* = 42) -endometriosis (*n* = 31) -male factor (*n* = 25) -unexplained (*n* = 39)	-PCOS (*n* = 69) -endometriosis (*n* = 56) -male factor (*n* = 25) -unexplained (*n* = 47)	0.2133
Total numbers of oocytes retrieved	7 (2–12)	5 (2–8)	2.2 × 10^−16^
Blastocysts transferred	2	2	-
Good quality embryos on day 3	2 (1–4)	2(1–4)	0.4

**Table 2 jcm-11-01217-t002:** Live birth in the study group according to the detected and corrected abnormally.

Corrected Abnormally	Live Birth Clinical Pregnancy	Corrected Abnormalities (Number of Cases)
Endometritis	65%	3
Intrauterine polyps	65%	19
Adhesions	80%	4
Submucosal fibroids	100%	13
Endocervical polyp	50%	2

**Table 3 jcm-11-01217-t003:** Proportion comparisons for all endpoints.

Variable	Study Group	Control Group	Difference	*p*-Value
No. of Live Birth ^1^	58/137 (42.3%)	52/197 (26.3%)	16%	0.0025
No. of Biochemical Pregnancy ^2^	4/137 (2.9%)	4/197 (2%)	0.9%	0.603
No of Miscarriage ^3^	30/137 (21.8%)	35/197 (17.7%)	4.1%	0.35

^1^ A birth at which a child is born alive. ^2^ A biochemical pregnancy is a very real pregnancy where implantation did occur but one that results in a miscarriage within the first 2–3 weeks of conception. ^3^ Spontaneous Loss of a pregnancy before the 20th week.

## Data Availability

Data analysis will include personal data which cannot be shared with public. However, if institutions or authorised individuals will require the data analysis then are welcome to email the team.

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
