# Peer review of "Is Hysteroscopy Prior to IVF Associated with an Increased Probability of Live Births in Patients with Normal Transvaginal Scan Findings after Their First Failed IVF Trial?"

_jcm, 2022, doi:10.3390/jcm11051217_

Round 1

Reviewer 1 Report

Thank you for your responses to the comments provided and there is significant improvement in the current manuscript. However, the methodology, results section, and conclusion still require improvement in order to demonstrate experimental soundness. For example, I am unclear if a regression analysis is the appropriate way to analyze the results provided and similarly am unclear with what inputs you placed into it (your dependent and independent values). 

Author Response

Dear Reviewer,

Thank you for spending time in reviewing this manuscript. Please find our answers below. 

Thank you for your responses to the comments provided and there is significant improvement in the current manuscript. However, the methodology, results section, and conclusion still require improvement in order to demonstrate experimental soundness. For example, I am unclear if a regression analysis is the appropriate way to analyze the results provided and similarly am unclear with what inputs you placed into it (your dependent and independent values). 

 Thank you for your review . The statistical analysis was conducted in programming language R. Given the fact that the aim of the study was  the impact of the hysteroscopy, a Generalised Linear Model (GLM) was used which is a class of regression model that supports non normal distributions ie whether hysteroscopy was performed (yes or no question). Therefore, as we had categorical values , GLM analysis with binomial family  was performed and the the p- value corresponding to the z statistic was used. Regarding the analysis in Table 1, the data was analysed  using t-test and Chi-squared. 

Overall, we tried to improve the manuscript and hope you will be satisfied. 

Reviewer 2 Report

I have looked again into the revised manuscript and I have also verified the changes that the authors (may) have introduced, based on my previous comments. It is my duty to convey to you my findings. Unfortunately,  the manuscript still needs to be improved.   I will provide you with a few examples:    

  1. Any differences in the main study characteristics (Table 1) were not analyzed statistically and the explanation that there are no differences cannot be accepted. Differences in the main study characteristics must be analyzed and not only differences, but also similarities must be quantified. Overall, the layout of the Table 1 remains rather sloppy. For example: instead of "total follicles retrieved" should be "total number of oocytes retrieved", etc.
  2. Chapter 2.4: It was asked to denominate the method of statistical analysis. Unfortunately, "regression analysis" is not the appropriate method and also the presentation of the results does not fit to a regression analysis.
  3. Chapter 2.2: it was asked to replace "Final maturation" with "ovulation induction". Unfortunately, the word "final" has not been omitted.
  4. Table 2: the three variables should be accompanied by the unit of measurement, i.e. number, etc.
  5. I could not find reference no. 19 in the manuscript.
  6. I could not understand by any means the statement in the discussion (lines 167...). If the authors failed to perform their study due to the pandemic, how can this lead to a better comparison of both groups? This statement remains unclear.

Author Response

Dear Reviewer,

Thank you for spending time in reviewing this manuscript. Please find our answers. 

  1. Any differences in the main study characteristics (Table 1) were not analyzed statistically and the explanation that there are no differences cannot be accepted. Differences in the main study characteristics must be analyzed and not only differences, but also similarities must be quantified. Overall, the layout of the Table 1 remains rather sloppy. For example: instead of "total follicles retrieved" should be "total number of oocytes retrieved", etc.- The differences in Table 1 have now been analysed statistically using t-test and Chi-squared. The table has also now been amended and we hope you will be satisfied and find it less sloppy. 
  2. Chapter 2.4: It was asked to denominate the method of statistical analysis. Unfortunately, "regression analysis" is not the appropriate method and also the presentation of the results does not fit to a regression analysis.- The statistical analysis was conducted in programming language R. Given the fact that the aim of the study was  the impact of the hysteroscopy, a Generalised Linear Model (GLM) was used which is a class of regression model that supports non normal distributions ie whether hysteroscopy was performed (yes or no question). Therefore, as we had categorical values , GLM analysis with binomial family  was performed and the the p- value corresponding to the z statistic was used. 
  3. Chapter 2.2: it was asked to replace "Final maturation" with "ovulation induction". Unfortunately, the word "final" has not been omitted.-It has been corrected
  4. Table 2: the three variables should be accompanied by the unit of measurement, i.e. number, etc. - It has been amended
  5. I could not find reference no. 19 in the manuscript. - It has been added
  6. I could not understand by any means the statement in the discussion (lines 167...). If the authors failed to perform their study due to the pandemic, how can this lead to a better comparison of both groups? This statement remains unclear.- We are sorry for that,It has now been corrected. it was not meant to be placed there. 

Round 2

Reviewer 1 Report

Significant improvement from prior drafts, and thank you for your careful changes to incorporate many comments and suggestions. Several minor points for improvement include: 

1) Abstract: the language used is a convoluted and confusing for the following sentences: "In this retrospective study we compared the probability of live birth between women who after the first failed IVF/ICSI attempt underwent a hysteroscopy and those who were evaluated by a transvaginal scan. The retrospective study took place at the Assisted Reproductive Unit of the University Hospital of Ioannina from 2017 to 2020. It included 334 women with normal findings in a repeat ultrasound scan after a failed IVF/ICSI trial, 137 of whom underwent in turn diagnostic hysteroscopy before the next IVF/ICSI." There are too many confusing clauses with those, whom, etc that make it difficult to follow. Please simplify the language. 

2) Introduction: Line 64 "We scheduled this retrospective study" is an awkward phrasing to describe how you planned to set up the project. Please change word scheduled. 

3) Material and Methods: Perhaps change the word "reason" in line 73 to "goal" or "aim". For describing the high quality embryos in line 78, it may be helpful to referring to or cite SART embryo grading or other embryo grading systems that are used similar grading definitions. Line 102 requires the word "with" prior to soft catheter. Line 117 - please spell out GLM model and cite. 

4) Results: Overall improvement. Please see table specific comments. Thank you for including section 3.3 as this is very enlightening. 

5) Table 1: a p value of 1.41 e 10^-15 is HIGHLY significant (age) as well as for BMI, duration, and number of oocytes. Is this an error for the p value or are these groups truly different? If they are different, please comment on the differences within the results/discussion. As well, add (n) next to "causes of infertility" within the table. 

6) Table 2: I would actually switch section 3.2 and 3.3 in order with each other so that you discuss hysteroscopic findings first. 

7) Table 3: Please include ns of each abnormality even though you placed it in the written text for both corrected abnormality and live birth clinical pregnancy 

8) Discussion/Conclusions: Significant Improvements!! Well done. 

Overall, significant improvement in this manuscript. 

Author Response

Dear Reviewer,

Thank you for taking the time to review this manuscript. Please find our corrections below. we Hope you will be satisfied with the result.

kind regards,

Apostolia Galani

1) Abstract: the language used is a convoluted and confusing for the following sentences: "In this retrospective study we compared the probability of live birth between women who after the first failed IVF/ICSI attempt underwent a hysteroscopy and those who were evaluated by a transvaginal scan. The retrospective study took place at the Assisted Reproductive Unit of the University Hospital of Ioannina from 2017 to 2020. It included 334 women with normal findings in a repeat ultrasound scan after a failed IVF/ICSI trial, 137 of whom underwent in turn diagnostic hysteroscopy before the next IVF/ICSI." There are too many confusing clauses with those, whom, etc that make it difficult to follow. Please simplify the language. - It has been amended and should be less confusing now. 

2) Introduction: Line 64 "We scheduled this retrospective study" is an awkward phrasing to describe how you planned to set up the project. Please change word scheduled. - It has been amended.

3) Material and Methods: Perhaps change the word "reason" in line 73 to "goal" or "aim". For describing the high quality embryos in line 78, it may be helpful to referring to or cite SART embryo grading or other embryo grading systems that are used similar grading definitions. Line 102 requires the word "with" prior to soft catheter. Line 117 - please spell out GLM model and cite. - Overall they have been amended. 

4) Results: Overall improvement. Please see table specific comments. Thank you for including section 3.3 as this is very enlightening. 

5) Table 1: a p value of 1.41 e 10^-15 is HIGHLY significant (age) as well as for BMI, duration, and number of oocytes. Is this an error for the p value or are these groups truly different? If they are different, please comment on the differences within the results/discussion. As well, add (n) next to "causes of infertility" within the table. -It has been amended. 

6) Table 2: I would actually switch section 3.2 and 3.3 in order with each other so that you discuss hysteroscopic findings first. -It has been amended.

7) Table 3: Please include ns of each abnormality even though you placed it in the written text for both corrected abnormality and live birth clinical pregnancy - extra column has been added

Reviewer 2 Report

The authors have now responded to all questions raised.

Author Response

Dear Reviewer,

Thank you for taking the time to review this manuscript. As you can see it has been revised again and we hope that you agree with the final changes. Once again, thank you for your comments and reviews.

Kind regards,

Apostolia Galani

This manuscript is a resubmission of an earlier submission. The following is a list of the peer review reports and author responses from that submission.

Round 1

Reviewer 1 Report

OVERVIEW

This is a retrospective study conducted in Europe on 334 women with normal repeat ultrasound scan following failed IVF/ICSI. The central question of the manuscript is whether the addition of diagnostic hysteroscopy in this cohort will improve the rates of live births. Overall, the manuscript focuses on an interesting and pertinent research questions surrounding implantation failure following ICSI/IVF without known intrauterine pathology. However, there are several weaknesses to the manuscript that limit enthusiasm on this work, but may be revisable.    

INTRODUCTION  

The introduction is overall well written with a clear research question and objective. There are several minor areas for improvement:  

  1. There are very few studies referenced from the past 10 years, and even fewer from the past 5 years. Are there any more recent articles pertaining to implantation failure within normal uterine cavity contour with hysteroscopy?  
  2. The discussion regarding sensitivity and specificity of imaging modalities is somewhat difficult to read and may be improved with clearer language. Additionally, it would be wise to include saline infuse sonogram as an additional methods for evaluation of the uterine cavity.    

METHODS  

The methods section is overall well written and provides a significant amount of clarity regarding the study. There are small aspects that can be clarified.  

  1. What statistical analysis was used to assess the data? This should be included within the methods section.  
  2. Please describe the specific ways the good quality embryos on day 3 are assessed. Specifically, which embryos were included as high-quality and which ones were not.  
  3. Was PGT-A available for analysis of embryos to determine euploidy?  
  4. Were there any reasons why the 137 people chosen for hysteroscopy? Was it based on provider preference? Was there something inherent to this group compared to the 197 comparison group. Please comment on this aspect of the groupings from the retrospective design.  
  5. Why was this specific inclusion/exclusion group chosen? Why was the age cut off 43 years old?  
  6. If there was an intervention during hysteroscopy, how long following procedure was standard for IVF and embryo transfer? Were all transfers fresh cycles?

RESULTS AND TABLES  

The results section is missing several important main results portrayed in an easily accessible manner. The table are confusing and do not supplement a reader’s understanding of the results in their present form. This section is the weakest portion of the paper and requires significant improvement in order to ultimately arrive at the author’s conclusion. Several keys points are:  

  1. For the 26 people excluded, which group were they a part of (skewed one why or another?)
  2. The authors report that their secondary outcome was abnormal hysteroscopic findings, however, there is no mention or report of the specific hysteroscopic findings, the n values seen (normal vs abnormal), number of resections, endometrial biopsies, etc.  
  3. Table 1 has several significant areas for improvement in order to understand the main study characteristics and whether the two groups were matched appropriately:  
    1.  There is a footnote to “a” (ages, BMI, total follicles, and good quality embryos) but no footnote assigned to this information  
    2. Please include p values for all demographic qualities to assess whether the groups are matched appropriate or comparable  
    3. Is the age (years) median and range? Why is there a 46 year old in the Group TVS if your exclusion criteria was <43 years old?  
    4. For BMI, please include whether this is median or mean and its associated range or st.dev  
    5. For duration of infertility, is this years of trying to conceived or years under infertility doctors care?
    6.  For causes of infertility, include all n values of each major cause  
    7. The blastocysts transferred, please supply the range that is included  
    8. For good quality embryos on day 3, please explain further what this means morphologically to be included.  
  4. Table 2 has several significant areas for improvement  
    1. Please include how you completed your statistics as when repeated your analysis, I obtained different p values for each of the variables in Table 2, including a nearly statistically significant different in miscarriage rate between the two groups.  
    2. It would be helpful for you to complete further subgroup analysis based on this information, such as splitting your groups into ages <35 and >35 years old and seeing whether there may be other factors influencing the differences in live birth rate (or include in supplemental analysis). As it stands at present, a singular significant p value is not enough to ultimately claim that hysteroscopy following failed first IVF/ISCI increased live birth rate.  
  5. Table 3 has significant areas for improvement  
    1. This table provides no additional information to the reader. Please include n values for hysteroscopy results as well as the number of people who did not have findings/resection at time of hysteroscopy. It may additionally be useful to provide the reader with analysis of those who underwent hysteroscopy with and without interventions (polyp, etc), whether those two groups had different outcomes.    

DISCUSSION/CONCLUSION

The discussion and conclusion of the paper makes a grand claim regarding the results that is not backed by the author’s results in its current form. That being said, the dicussion is overall well written and explained. Several key aspects to add would include:

  1. The authors should discuss limitations of the specific study including confounding and bias with their specific studies.
  2. Future directions should be discussed in the discussion/conclusion (at present, there is mention in the abstract for future directions that are not discussed in the body of the paper).         

Author Response

The introduction is overall well written with a clear research question and objective. There are several minor areas for improvement:  

  1. There are very few studies referenced from the past 10 years, and even fewer from the past 5 years. Are there any more recent articles pertaining to implantation failure within normal uterine cavity contour with hysteroscopy?        -Thank you for your comment. We have included those studies that were relevant however it is true that we have missed some of those. We have modified the references accordingly. 
  2. The discussion regarding sensitivity and specificity of imaging modalities is somewhat difficult to read and may be improved with clearer language. Additionally, it would be wise to include saline infuse sonogram as an additional methods for evaluation of the uterine cavity.   
  3.           -Thank you for your review. The discussion regarding sensitivity and specificity of imaging modalities has been modified accordingly and furthermore, saline infusion sonogram has also been added.  
  4.  

METHODS  

The methods section is overall well written and provides a significant amount of clarity regarding the study. There are small aspects that can be clarified.  

  1. What statistical analysis was used to assess the data? This should be included within the methods section.  - Thank you for your comment. The statistical analysis that was used was regression analysis and it has been added to the methods section now. 
  2. Please describe the specific ways the good quality embryos on day 3 are assessed. Specifically, which embryos were included as high-quality and which ones were not.  -Embryo scoring on peimplantation development day 3 stratified the embryos in categories A,B,C, and D, based on their morphology. Category A included embryos with evenly cleaved blastomeres (even blastomere sizes), without any cellular fragments present. Category B included embryos with evenly cleaved blastomeres, with small amounts of cellular fragments present in less than 25% of the total embryo volume. Category C included embryos with uneven blastomeres sizes, and/or cytoplasmic debris or cellular fragments present in up to 50% of the total embryo volume. Category D included embryos with uneven blastomeres sizes, and/or cytoplasmic debris or cellular fragments present in more than 50% of the total embryo volume. Categories A and B are the assesed as good quality embryos on day 3.
  3. Was PGT-A available for analysis of embryos to determine euploidy? -PGT-A is rutinely performed in our lab with permission from the Greek Authority for Medically Assisted Reproduction for every individual couple with particular medical indications for PGT-A . The indications for the patients recruitment in this study were not consistent with the scope of this study and therefore not applicable to the patients stratified for this study. 
  4. Were there any reasons why the 137 people chosen for hysteroscopy? Was it based on provider preference? Was there something inherent to this group compared to the 197 comparison group. Please comment on this aspect of the groupings from the retrospective design. - Our initial goal was both of the groups to consist of the same amount of participants. However , due to COVID-19 pandemic there has been significant reduction in the patients who could undergo IVF/ICSI funded by the National Health System. Therefore, as our milestone was to complete the study in 3 years , we included the participants that were gathered so far.
  5. Why was this specific inclusion/exclusion group chosen? Why was the age cut off 43 years old?  - Inclusion and exclusion group was determined according to our Trust guidelines. In the most national guidelines, the age limit is 42-45 years old. According to this Trust’s guidelines the cut off age for IVF/ICSI is 43 years old. Therefore, we included all the women up to 43 years. 
  6. If there was an intervention during hysteroscopy, how long following procedure was standard for IVF and embryo transfer? Were all transfers fresh cycles?- For all the cases that an intervention took place, we waited for one cycle and then we proceeded with the IVF/ICSI. 

RESULTS AND TABLES  

The results section is missing several important main results portrayed in an easily accessible manner. The table are confusing and do not supplement a reader’s understanding of the results in their present form. This section is the weakest portion of the paper and requires significant improvement in order to ultimately arrive at the author’s conclusion. Several keys points are:  

  1. For the 26 people excluded, which group were they a part of (skewed one why or another?) -10 were from the study group and 16 from the control group. Has been added.
  2. The authors report that their secondary outcome was abnormal hysteroscopic findings, however, there is no mention or report of the specific hysteroscopic findings, the n values seen (normal vs abnormal), number of resections, endometrial biopsies, etc.  -Table 3 includes the abnormalities. 
  3. Table 1 has several significant areas for improvement in order to understand the main study characteristics and whether the two groups were matched appropriately:  
    1.  There is a footnote to “a” (ages, BMI, total follicles, and good quality embryos) but no footnote assigned to this information  -It has been corrected 
    2. Please include p values for all demographic qualities to assess whether the groups are matched appropriate or comparable  -Our questionnaires in the two study groups were anonymized  as they requested personal data and therefore at this point to retrieve the data it may require significant deadline extension. 
    3. Is the age (years) median and range? Why is there a 46 year old in the Group TVS if your exclusion criteria was <43 years old? -Thank you for this note. It has been corrected. 
    4. For BMI, please include whether this is median or mean and its associated range or st.dev -It has been modified
    5. For duration of infertility, is this years of trying to conceived or years under infertility doctors care?-It is years trying to conceive, it has ben modified
    6.  For causes of infertility, include all n values of each major cause  
    7. The blastocysts transferred, please supply the range that is included - As mentioned in table 1, the blastocysts transferred for every patient were 2.   
    8. For good quality embryos on day 3, please explain further what this means morphologically to be included. -Good morphology blastocysts grade A and B with well formed ICM.
    9. Table 2 has several significant areas for improvement  
    10. Please include how you completed your statistics as when repeated your analysis, I obtained different p values for each of the variables in Table 2, including a nearly statistically significant different in miscarriage rate between the two groups. -The data in this study were processed with regression analysis.  
    11. It would be helpful for you to complete further subgroup analysis based on this information, such as splitting your groups into ages <35 and >35 years old and seeing whether there may be other factors influencing the differences in live birth rate (or include in supplemental analysis). As it stands at present, a singular significant p value is not enough to ultimately claim that hysteroscopy following failed first IVF/ISCI increased live birth rate. - Thank you for your comment. It would indeed be helpful. However, please note that this specific study concentrated on the use of hysteroscopy after a failed IVF/ICSI. Therefore, I wonder if we proceeded with subgroup analyses specifically with age it would focus on a different aspect.
  4. Table 3 has significant areas for improvement  
    1. This table provides no additional information to the reader. Please include n values for hysteroscopy results as well as the number of people who did not have findings/resection at time of hysteroscopy. It may additionally be useful to provide the reader with analysis of those who underwent hysteroscopy with and without interventions (polyp, etc), whether those two groups had different outcomes.  - We have mentioned that 30% of the study group had abnormal findings. We have added in the results the number of patients who had resection and treatment.  

DISCUSSION/CONCLUSION

The discussion and conclusion of the paper makes a grand claim regarding the results that is not backed by the author’s results in its current form. That being said, the dicussion is overall well written and explained. Several key aspects to add would include:

  1. The authors should discuss limitations of the specific study including confounding and bias with their specific studies.-It has been modified.
  2. Future directions should be discussed in the discussion/conclusion (at present, there is mention in the abstract for future directions that are not discussed in the body of the paper).  -It has been modified.   

Reviewer 2 Report

In this paper entitled "Is hysteroscopy prior to IVF associated with an increased probability of live births in patients with normal transvaginal scan findings after their first failed IVF trial?" the authors retrospectively reviewed the patient who fails her first cycle. As stressed in this article, this patient often has no obvious etiology and a normal recent ultrasound. Yet, they noted in the review of over 300 of their own patients that 30% actually had underlying uterine anomalies not seen on regular ultrasound. In those who underwent a hysteroscopy, improvement in live birth was noted. 

There are several strengths of this study, most notable is that this type of patient is all too common. Additionally, they broke down the type of anomaly noted and assessed live birth rate post treatment. 

Areas of improvement:

  1. Overall- this topic is not new, Farahat et al in 2014, Al-Temary et al in 2019, and Okohue et al in 2020 all reported improvement in outcomes when hysteroscopy was utilized after at least one failed IVF transfer. Stress the areas of your own study that are different and adds to the already robust area of literature.
  2. The abstract states that abnormal endometrial findings were seen in 30% of patients (line 28), however this is not in the body of the paper.
  3. Define what was characterized as 'high quality embryo' (line 75)
  4. Materials and Methods:
    1. Consider power calculation of how many patients would be needed to see an x% difference in live birth rate to determine effect of change.
    2. Break down how many were day 3 versus day 5 transfers in each group, as outcomes differ between the two- how many of each were in the study versus control group? (Line 98).
  5. Results
    1. Table 1: are any of the differences significant?
    2. In the Group TVS- age range was up to 46, although you stated in exclusion criteria that the age limit was 43.
    3. Total follicles- is this at retrieved or seen on baseline ultrasound?
    4. Break down day 3 and day 5 transfers
  6. Discussion
    1. Need to make it more of a robust overall discussion- include strength and weakness of your study and state what makes your study different than the several others that reported improvement in outcome after hysteroscopy. 
  7. Conclusion
    1. Does not match your original hypothesis- hypothesis stated hysteroscopy after failed transfer, while your conclusion states before IVF/ICSI.

Author Response

Dear Reviewer,

Thank you for your valuable input in our manuscript. Please find below our answers to your questions.

  1. Overall- this topic is not new, Farahat et al in 2014, Al-Temary et al in 2019, and Okohue et al in 2020 all reported improvement in outcomes when hysteroscopy was utilized after at least one failed IVF transfer. Stress the areas of your own study that are different and adds to the already robust area of literature.-In this study, the patients underwent a hysteroscopy after the first failed IVF/ICSI as compared to the aforementioned studies where the patients had a hysteroscopy in most cases after two previous  failed IVF cycles. Having a hysteroscopy after one failed IVF/ICSI attempt and subsequently identifying pathology that could affect the implantation, could lead to minimise the patients stress levels and reduce the extra cost that comes with it. Moreover, the groups size of this study is bigger compared to the studies mentioned and therefore the outcome is more reliable.
  2. The abstract states that abnormal endometrial findings were seen in 30% of patients (line 28), however this is not in the body of the paper. -It has been added.
  3. Define what was characterized as 'high quality embryo' (line 75)-High  quality embryo are those defined  as good quality embryos on day 3as well as those defined as good morhology blastocysts grade A and B . It has been added in the manuscript as well.
  4. Materials and Methods:
    1. Consider power calculation of how many patients would be needed to see an x% difference in live birth rate to determine effect of change.-although this is an excellent point, I believe that it would be more appropriate with equal size groups and it could confuse the reader. 
    2. Break down how many were day 3 versus day 5 transfers in each group, as outcomes differ between the two- how many of each were in the study versus control group? (Line 98).-There has been a mistake in typing. Although our original though was to transfer on day 3 and day 5, team has decided to carry on with embryo transfer only on day 3 so that the outcome will not be affected.It has been corrected in the manuscript as well.
  5. Results
    1. Table 1: are any of the differences significant?- Although there are no significant differences however the table 1 is important as it shows that both groups had similar characteristics. 
    2. In the Group TVS- age range was up to 46, although you stated in exclusion criteria that the age limit was 43. -It has been corrected
    3. Total follicles- is this at retrieved or seen on baseline ultrasound? -It refers to follicles retrieved
    4. Break down day 3 and day 5 transfers -Transfers only on day 3
  6. Discussion
    1. Need to make it more of a robust overall discussion- include strength and weakness of your study and state what makes your study different than the several others that reported improvement in outcome after hysteroscopy. -It has been modified. 
  7. Conclusion
    1. Does not match your original hypothesis- hypothesis stated hysteroscopy after failed transfer, while your conclusion states before IVF/ICSI. -Hysteroscopy is performed after a failed IVF/ICSI and prior to the next attempt. It has been modified. 

Reviewer 3 Report

The authors of this potential contribution to the Journal of Clinical Medicine present the results of a retrospective analysis of a cohort of 137 infertile women that underwent a diagnostic hysteroscopy after a first failed attempt of IVF. The control group consisted of a group of 197 infertile women that underwent a simple transvaginal ultrasound scan prior to their second attempt. The results indicate a significantly higher live birth rate in the study group as compared to the control group (p=0.025). The topic of this contribution is relevant and up to date, but the presentation of the data, the procedures and the outcomes in the three tables should be improved.

The methods of statistical analysis have not been described in the Material and Methods section.

Table 1: the number of causes of infertility in each of both study groups should be given. Unless all patients invariably received two blastocysts, the mean number of blastocysts should be accompanied by either min-max figures or by the standard of deviation of the number of blastocysts transferred. The same goes for the BMI.

Table 1 should also contain the statistical analysis of potential differences between both study groups.

Table 2: what is meant by “biochemical pregnancy”? Is it a positive pregnancy test? If so, the authors should provide us with the criteria of a positive pregnancy test in their institution. Why not enter “ongoing pregnancy”?

Table 3: the authors should include the absolute numbers of corrected abnormalities. In the text the authors should also provide us with the respective methodologies, such as the antibiotic(s) used to correct the diagnosed abnormalities, the dosages and the duration of treatment.

According to the manuscript, uterine cavity abnormalities were exclusively detected in the hysteroscopy group, whereas not a single abnormality was mentioned in the transvaginal ultrasound group, which consisted of 197 women. This appears unlikely. Please explain and, if needed, amend.

I also wonder why in the exclusion criteria the following were not included: uterine fibroids, history of hysteroscopy.

One important reference in the Cochrane database by Bosteels et al. (2018) should be included as well. Other relevant and recent meta-analyses are that by Cao et al. (2018) and that by Busnelli et al. (2021). Discussion, line 167: the TROPHY trial by El-Toukhy (2016) was not cited, neither in the discussion nor in the reference list.

Minor comments:

Introduction, page 1, line 41: «The probability of a pregnancy achievement is approximately 30%”: the percentage must be set in perspective, such as per treatment trial?

Page 2, lines 43-44: If the authors state that there is evidence to suggest that these may exert a negative impact on the likelihood of pregnancy, then they should provide us with the references.

Page 2: line 57: reference missing.

Page 2, lines 58-59: the authors list some maternal causes for failed implantation, but, again, appropriate references are lacking. I also doubt, whether there are immunological causes for failed implantation and thrombophilia may also be doubtful.

Page 2, line 89: what were considered as signs of endometritis? Various symptoms have been described, but proof of evidence is lacking. The authors should list the symptoms, that they followed, and provide the references.

Page 3, line 97: the term “final oocyte maturation” is wrong and should be replaced by ovulation induction with hCG.

Section 2.3: the authors should provide some information about the number of embryos that were transferred.

Page 3, line 109: please replace “dicomented” by “documented” . The authors should also expand on the treatments that were offered, if in hysteroscopy abnormal findings were recorded. They should also clarifiy on the diagnosis of “endometritis” in the endometrial biopsies including the appropriate reference(s).

Table 1: the parameters “causes of infertility”, “total follicles”, “blastocysts transferred” and “good quality embryos” should be accompanied by the item, which is “numbers” or “no”.

Discussion, line 134: second attempt, please correct.

Discussion, line 137: reference lacking.

Discussion, line 139: please correct “considering”.

Author Response

Dear Reviewer,

Thank you for your input. Please find below the answers to your questions.

The authors of this potential contribution to the Journal of Clinical Medicine present the results of a retrospective analysis of a cohort of 137 infertile women that underwent a diagnostic hysteroscopy after a first failed attempt of IVF. The control group consisted of a group of 197 infertile women that underwent a simple transvaginal ultrasound scan prior to their second attempt. The results indicate a significantly higher live birth rate in the study group as compared to the control group (p=0.025). The topic of this contribution is relevant and up to date, but the presentation of the data, the procedures and the outcomes in the three tables should be improved.

The methods of statistical analysis have not been described in the Material and Methods section.-It has been added.

Table 1: the number of causes of infertility in each of both study groups should be given. Unless all patients invariably received two blastocysts, the mean number of blastocysts should be accompanied by either min-max figures or by the standard of deviation of the number of blastocysts transferred. The same goes for the BMI.

Table 1 should also contain the statistical analysis of potential differences between both study groups. -Patients invariably have received two blastocysts, BMI also clarified. Overall as can be seen the differences in table 1 are not significant and therefore we did not proceed with further statistical analysis. 

Table 2: what is meant by “biochemical pregnancy”? Is it a positive pregnancy test? If so, the authors should provide us with the criteria of a positive pregnancy test in their institution. Why not enter “ongoing pregnancy”?- A biochemical pregnancy is a very real pregnancy where implantation did occur but one that results in a miscarriage within the first 2-3 weeks of conception. The definitions  have been added in the manuscript. Biochemical pregnancy has not been added as an ongoing pregnancy as it does not last more than 3 weeks and may have false positive pregnancy test. 

Table 3: the authors should include the absolute numbers of corrected abnormalities. In the text the authors should also provide us with the respective methodologies, such as the antibiotic(s) used to correct the diagnosed abnormalities, the dosages and the duration of treatment.-It has been added.

According to the manuscript, uterine cavity abnormalities were exclusively detected in the hysteroscopy group, whereas not a single abnormality was mentioned in the transvaginal ultrasound group, which consisted of 197 women. This appears unlikely. Please explain and, if needed, amend.-Indeed, 52 patient has thickened endometrium which could be an endometrial polyp or submucosal fibroid. It has been amended.

I also wonder why in the exclusion criteria the following were not included: uterine fibroids, history of hysteroscopy. - A hx of a previous hysteroscopy would not be in the exclusion criteria as pathology such as polyps, endometritis etc could happen at any point during reproductive age. Fibroids that do not affect the endometrium, would not affect the implantation and therefore, they should not be in the exclusion criteria.

One important reference in the Cochrane database by Bosteels et al. (2018) should be included as well. Other relevant and recent meta-analyses are that by Cao et al. (2018) and that by Busnelli et al. (2021). Discussion, line 167: the TROPHY trial by El-Toukhy (2016) was not cited, neither in the discussion nor in the reference list. -It has been amended. 

Minor comments:

Introduction, page 1, line 41: «The probability of a pregnancy achievement is approximately 30%”: the percentage must be set in perspective, such as per treatment trial?

Page 2, lines 43-44: If the authors state that there is evidence to suggest that these may exert a negative impact on the likelihood of pregnancy, then they should provide us with the references. -It has been amended. 

Page 2: line 57: reference missing.- It has 2 references.

Page 2, lines 58-59: the authors list some maternal causes for failed implantation, but, again, appropriate references are lacking. I also doubt, whether there are immunological causes for failed implantation and thrombophilia may also be doubtful.-Amended

Page 2, line 89: what were considered as signs of endometritis? Various symptoms have been described, but proof of evidence is lacking. The authors should list the symptoms, that they followed, and provide the references.- endometritis can be asymptomatic or give symptoms such as abnormal vaginal bleeding, swelling of the abdomen, general discomfort etc. In hysteroscopy the findings usually are micro polyps, stroll edema. However the diagnosis and the suspicion is also based on experience. Not sure however if references and expanding of findings are relevant there.

Page 3, line 97: the term “final oocyte maturation” is wrong and should be replaced by ovulation induction with hCG. -Amended.

Section 2.3: the authors should provide some information about the number of embryos that were transferred. -Amended.

Page 3, line 109: please replace “dicomented” by “documented” . The authors should also expand on the treatments that were offered, if in hysteroscopy abnormal findings were recorded. They should also clarifiy on the diagnosis of “endometritis” in the endometrial biopsies including the appropriate reference(s). -Amended

Table 1: the parameters “causes of infertility”, “total follicles”, “blastocysts transferred” and “good quality embryos” should be accompanied by the item, which is “numbers” or “no”. -Amended

Discussion, line 134: second attempt, please correct.

Discussion, line 137: reference lacking.

Discussion, line 139: please correct “considering”.